# Multi-Focus Image Fusion Based on Multi-Scale Generative Adversarial Network

**DOI:** 10.3390/e24050582

**Published:** 2022-04-21

**Authors:** Xiaole Ma, Zhihai Wang, Shaohai Hu, Shichao Kan

**Affiliations:** 1School of Computer and Information Technology, Beijing Jiaotong University, Beijing 100044, China; maxiaole@bjtu.edu.cn (X.M.); zhhwang@bjtu.edu.cn (Z.W.); 2Beijing Key Laboratory of Advanced Information Science and Network Technology, Beijing 100044, China; 3School of Computer Science and Engineering, Central South University, Changsha 410083, China; kanshichao@csu.edu.cn

**Keywords:** multi-scale decomposition, generative adversarial network, multi-focus image fusion

## Abstract

The methods based on the convolutional neural network have demonstrated its powerful information integration ability in image fusion. However, most of the existing methods based on neural networks are only applied to a part of the fusion process. In this paper, an end-to-end multi-focus image fusion method based on a multi-scale generative adversarial network (MsGAN) is proposed that makes full use of image features by a combination of multi-scale decomposition with a convolutional neural network. Extensive qualitative and quantitative experiments on the synthetic and Lytro datasets demonstrated the effectiveness and superiority of the proposed MsGAN compared to the state-of-the-art multi-focus image fusion methods.

## 1. Introduction

In image collection, many sensors have been developed to acquire different types of images; however, due to the limitations of the application scenarios and working environment, they usually have different specific functions, leading to some limitations in the collected information. For example, due to the limitation of the sensors with different focal lengths and the scenes with different depths of field, the degree of focus of the scene reflected by the image is also different, as shown in Figure 1a,b. In order to make full use of the complementarity and overcome the limitations of acquired images, image fusion can fully integrate a large amount of image information, obtain intuitive and concise image descriptions, and provide more a diversified information reference for target detection and recognition [1,2].

The initial image fusion was mainly applied in the fields of military security and Earth remote sensing. For example, the fusion of infrared and visible images can improve the recognition rate of targets. In addition, the employed intuitive algebraic methods are relatively simple, and have the characteristics of simple structure, fast calculation speed, and high execution efficiency. However, due to the inability to extract information in the frequency domain, it is easy to cause image degradation such as distortion and blocking effects [3]. As a result of the proposal and continuous improvement of the wavelet transform, image fusion methods based on multi-scale transform (MST) have been favored by many scholars [4,5], which achieve image fusion by converting source images from the spatial domain to the frequency domain for processing. However, because a fixed orthogonal basis function is used to transform the image, it is difficult to completely and effectively represent the rich features contained in the image. As a special kind of image transformation, sparse representation abandons the fixed orthogonal bases, and uses a redundant dictionary to realize the transformation of image information through approximately linear representations; however, the accompanying redundant dictionary training and sparse coding require a large number of calculations and consume more time, and it is difficult to ensure the timeliness of the method [6].

In the traditional image fusion methods, it is necessary to manually design and adjust the activity level measurement method and the fusion rule, of which the realization process is relatively complicated and the size of the calculation is large. As a result of the rapid development of machine learning, especially deep learning, the convolutional neural network (CNN) has also achieved good results in recent years. The image fusion methods based on deep learning can automatically extract the features of the source image by constructing CNN models of different depths; then, under the constraint of the loss function, the ideal fused image is finally output after multiple training and optimization. Once the network structure model is built, the entire fusion process does not require human involvement, and the network can automatically extract image features and directly output the fusion results. Thus, it is widely used in various image fusion approaches [7,8].

Liu first proposed the use of convolutional neural networks in image fusion tasks, using image blocks with different degrees of blur as the input images of the training network, and a CNN to extract the feature activity level and design fusion rules, which can be regarded as classification tasks; this avoided the need to manually design complex fusion strategies [7]. Prabhakar proposed the use of a deep unsupervised fusion model for multi-exposure image fusion, and achieved good performance in the fusion result by the unreferenced image metric, that is, structural similarity as the loss function, and the twin network structure as the feature extraction network [9]. Tang proposed a pixel convolutional neural network where the training data was generated by adding a defocus mask to the images, and the proposed method reduced the time complexity of the fusion task and further improved the fusion performance by dividing the pixels in source images into three categories: focus, defocus, and unknown [10]. Although the aforementioned methods have achieved good fusion effects, they still have some shortcomings. For example, some convolutional neural network methods do not make full use of the features of the middle layer of the network, and the effectiveness of the training results cannot be guaranteed when the amount of training data is small. In order to make full use of the features of the middle layer, Li proposed the use of the pre-trained VGG19 model as a tool for high-level feature extraction and achieved a better fusion effect [11]; however, there were many parameters and a large amount of calculation in the pre-trained model.

Following the proposal of the generative adversarial network (GAN) [12], it has been widely used in computer vision fields such as image generation and super-resolution reconstruction. Ma first proposed infrared and visible image fusion based on the generative adversarial network, where the images superimposed by the channels were input into the generator and the fused image obtained by the generator and the visible image were input into the discriminator. This method can ensure the fused image obtains more details, and achieved good results in the fusion of infrared and visible images by alternately cooperating and updating the parameters of the generator and the discriminator [13]. However, the network structure is relatively simple, and the fusion effect is greatly reduced when the type of source image changes, especially when the source images are multi-focus images with the same modality. Guo proposed a more complex conditional generative adversarial network for multi-focus image fusion; this model obtained the final fused image by post-processing the source images and the focus point confidence map generated by the generator [14].

The image fusion method based on GAN can further solve the problems of insufficient clarity of the fused image and weak correlation with the source images. Then, the depth of the network is usually increased to achieve a better extraction of high-level details. However, the deepening of the network depth does not always result in the improvement in network performance; on the contrary, a network that is too deep will have problems, such as gradient disappearance and explosion, which result in slower convergence and lower accuracy. Therefore, before processing the images by GAN, multi-scale decomposition is employed to obtain sub-images with more detailed information in different frequencies, and network training is then performed on these sub-images to alleviate the contradiction between network depth and feature extraction. The results of subjective evaluation and objective metrics of fused images in the compared experiments strongly verify the effectiveness of the proposed method. By comparing the images fused by ECNN and MsCNN, the increased values of image feature-based metrics ($Q_{G}$ and $Q_{P}$) demonstrate that the introduction of multi-scale decomposition improves the ability of the network to extract image features. The contributions of this paper can be summarized as follows:

(1) As the depth of the network increases, the extracted image features are more advanced, and the obtained image processing results are better. However, the increase in network depth does not always produce ideal processing results: when the network level reaches a certain level, the network performance not only does not improve, but may degenerate, and the computational complexity and the number of parameters will also greatly increase. The proposed method in this paper improves the ability of image feature extraction by multi-scale decomposition without increasing the network depth at the expense of network performance.

(2) Most image fusion methods based on neural networks are only applied to a part of the image fusion process, such as feature extraction, and the extracted features still need to manually measure the activity level and design fusion rules. Using a GAN with a different depth of network structure, the proposed method not only avoids manual operation, reduces complexity, and improves the universality of the method, but also makes full use of the information in different sub-images with rich features.

(3) For image fusion, “standard, well-fused” images are needed to be solved and do not exist in the real sense. Furthermore, due to the limitations of image acquisition and registration technology, it is very challenging to collect ideal training samples. A synthetic multi-focus image dataset is constructed in this paper by adding random blur kernels to different objects in the images of the image segmentation dataset, which overcomes the difficulties of the small number of training samples, poor registration accuracy of source images, and the lack of “standard, well-fused” reference images.

The remainder of this paper is arranged as follows: the related work is introduced in Section 2, including some related image fusion methods in Section 2.1 and the description of image fusion based on the GAN. The flowchart and more details about the proposed method are presented in Section 3, and Section 4 compares the experimental results and presents the analysis. Finally, the conclusion is presented in Section 5.

## 2. Related Work

A description of related work is presented in this section, including the introduction of the existing image fusion methods, the basic theory of the GAN, and its application in image fusion.

### 2.1. Image Fusion Methods

In general, image fusion can be divided into pixel-level, feature-level, and decision-level fusion [15], where pixel-level fusion is the basis for the other two approaches. In addition, there are three kinds of image fusion at the pixel level: image fusion based on the spatial domain, image fusion based on the transform domain, and image fusion based on deep learning.

Early image fusion was mostly based on the spatial domain, in which pixels were directly fused using fusion rules [16]. Due to the spatial domain processing of source images, less information is lost in image fusion. Low computational complexity, low computer performance requirements, and good real-time performance have also led to this kind of method being favored. Typical representatives of this approach are methods based on the weighted average and methods based on the maximization of the absolute value. The former method is very sensitive to the large difference in image pixels, whereas the images fused using the latter method have a strong sense of splicing and an incomplete structure. Furthermore, they are highly sensitive to noise and misregistration.

As a result of the development and wide application of image transformation, the methods based on the transform domain have good time-frequency characteristics, and can avoid problems in image fusion, such as the block effect and spectral distortion. There are three steps in the classical fusion framework of methods based on the transform domain: the choice of proper image transformation, the design of perfect fusion rules, and performing corresponding inverse transformation. For example, Wang proposed the multi-modal medical image fusion by Laplacian pyramid and adaptive sparse representation, where the LP decomposition was used to split medical images into four images of different sizes and adaptive sparse representation was performed to fuse the decomposed four layers; finally, the fused image was obtained by the inverse Laplace pyramid transform [17]. Chen proposed a target-enhanced multiscale transform (MST) decomposition model for infrared and visible image fusion to simultaneously enhance the thermal target in infrared images and preserve the texture details in visible images, where the decomposed infrared low-frequency information was used to determine the fusion weight of low-frequency bands and highlight the target, and a regularization parameter was introduced to dominate the proportion of the infrared features in a gentle manner [18]. As a result, designing an appropriate fusion rule for coefficients is a challenging prerequisite for good fusion results.

Compared with the aforementioned traditional image fusion methods, the methods based on convolutional neural networks have more convolution kernels that can be used to extract useful feature information (also a salient feature) of image depth, and can adaptively learn convolution parameters and complete parameter optimization by end-to-end training. For multi-focus image fusion, the existing deep fusion methods divide source images into small patches and apply a classifier to judge whether the patch is in focus. In contrast, Li proposed a novel deep network named deep regression pair learning (DRPL), which directly converts the whole image into a binary mask without any patch operation, subsequently tackling the difficulty of the blur level estimation around the focused/defocused boundary [19]. Based on this, the subsequent image fusion is mostly based on an end-to-end network, thus avoiding the cumbersome intermediate processing of the output.

### 2.2. Image Fusion Method Based on GAN

The generative adversarial network has been a popular research topic in recent years, and has achieved the best performance in tasks of image generation [20,21,22,23], such as image super-resolution reconstruction, image restoration, and style transfer. Drawing lessons from the “mini-max two-player game” in game theory, the GAN is composed of two competing neural networks: the discriminant network and the generative network. Among these, the generating network tries to generate data close to the real data, while the discriminating network tries to distinguish between the real data and the data generated by the generating network. During the confrontation between the two networks, the generating network uses the discriminant network as a loss function and updates its parameters to generate more real-looking data; in contrast, the discriminant network updates its parameters to better identify fake data from real data. This model can be modeled as (1).
(1)$$\underset{G}{\min}\underset{D}{\max}V(D,G)=E_{x\sim P_{data}(x)}\left[\log D(x)\right]+E_{z\sim P_{z}(z)}\left[\log\left(1-D\left(G(z)\right)\right)\right].$$
where $P_{data}$ denotes the true data distribution, $P_{z}(z)$ denotes the prior probability of training samples or random noise, G denotes the generative network, and D denotes the discriminant network. The output of the discriminant network is the probability value of the input close to the real data, and the value range is [0,1]. Furthermore, the optimization functions of the generating network and the discriminant network are shown in (2) and (3), respectively.
(2)$$\underset{G}{\min}V(D,G)=E_{z\sim P_{z}(z)}\left[\log\left(1-D\left(G(z)\right)\right)\right].$$
(3)$$\underset{D}{\max}V(D,G)=E_{x\sim P_{data}(x)}\left[\log D(x)\right]+E_{z\sim P_{z}(z)}\left[\log\left(1-D\left(G(z)\right)\right)\right].$$

However, due to the complexity of generating adversarial ideas and the difficulty in image training, its application in image fusion has just begun. Among these applications, the idea of generating adversarial training is to improve the quality of the fused image based on the generative and adversarial game strategies. If the quality of the generated image is poor, the loss of the generator is large, and the generator will be improved to generate higher quality images after calculating and feeding the loss back to it; otherwise, the loss of the discriminator is large, and its discrimination ability will be improved after the loss is fed back.

The framework of image fusion based on a GAN is shown in Figure 2, including the training and testing parts. The training part requires an adversarial game between the generator and the discriminator, whereas the testing part only needs the participation of the generator. In Figure 2, the number of images to be fused is M.

## 3. The Proposed Method

An effective and universal image fusion method based on a multi-scale generative adversarial network is proposed in this paper, which abandons the manual measurement of the activity level and the design of fusion rules in traditional image fusion methods, as shown in Figure 3. For the multi-scale image decomposition, the adoption of different filters and decomposition levels for an image can extract different features of the image and obtain sub-images of different numbers and sizes. In Figure 3, based on wavelet decomposition, the two source images are first decomposed into two low-frequency sub-images and multiple high-frequency sub-images. The low-frequency sub-images retain most of the content information of the image, while different high-frequency sub-images retain different levels of noise and detailed information. Taking into account the different richness of image details under different frequencies, different depth networks are designed to integrate these sub-images. Finally, based on the fusion results at various scales, the final fused image can be reconstructed by conducting the inverse wavelet transform. The steps of the proposed multi-focus image fusion method can be summarized as follows:

Step 1: Perform multi-scale decomposition on the source images, and transform them from the spatial domain to different frequency domains to obtain multi-scale sub-images;

Step 2: Considering that the richness of the detailed information in the sub-images at different frequencies is different, GANs with different depth network structures are used to fuse them; the network structure of the GAN is shown in Figure 4.

Step 3: Obtain the fused image by multi-scale reconstruction.

Here, for the convenience of display, the default number of source images is 2, and the decomposition scale is N.

### 3.1. Multi-Scale Generative Adversarial Network

For the proposed multi-scale GAN, a generator and a discriminator are designed for the fusion of the source images at each scale, as shown in Figure 4. The generator consists of two encoder modules: one feature fusion module, where Concat(f1, f2) means the fusion of Feature 1 with the blue color in Figure 4 and Feature 2 with the green color in Figure 4; and a decoder module, where ConvT is the abbreviation of ConvTranspose2d and “*2” means that there are two parallel decoder branches with the same construction, as shown in Table 1. Firstly, the sub-images of different source images are input to the encoder and different features are obtained; then, the feature maps are superimposed and input into the fusion network to obtain the fused feature map. After further processing of the fused feature map through the decoder, the fused sub-image can finally be obtained. In addition, it is also necessary to have a discriminator to assist in training for the generator, and the discriminator is separately designed for each generator. The input of the discriminator is an all-in-focus sub-image or a fused sub-image that is output by the decoder, and the main purpose is to determine if the input sub-image is generated or true.

In Figure 4, the encoder and feature fusion modules in the generator are constructed in a multi-layer convolution, and the decoder is built in a manner that corresponds to the multi-layer de-convolution corresponding to the generator. At the same time, in the encoder and decoder, drawing on the idea of the U-Net structure, each layer of the two encoders is connected with the corresponding layer of the decoder in a U-shaped jump connection, which achieves the full integration of information and decodes higher-quality images. The discriminator is directly established by the commonly used classification network, that is, it is realized via multiple convolutional layers and a fully connected layer classifier.

In the proposed sub-image fusion method, different depth encoders and discriminators for the sub-images at each decomposed scale are designed to achieve a more efficient fusion effect. Specifically, the sub-images obtained by different decompositions of an image contain diverse information. Therefore, the network of the same structure is difficult to be universally used for sub-images in end-to-end training. If a network with the same structure is used to encode and decode different sub-images, it will not only reduce efficiency, but also affect the network performance. When the decomposition scale is small, the sub-image contains more low-level information, such as texture, and deeper generators and discriminators need to be designed to better extract the deep information of sub-images; whereas, when the decomposition scale is large, generators and discriminators with shallow depth are needed to reduce unnecessary repetitive feature extraction operations. Finally, all sub-networks are combined into a multi-scale generative adversarial network to realize end-to-end training, so that the fused image is directly output when the source images are input.

### 3.2. Loss Function

Balancing loss and parameter optimization between the generation and adversarial networks is always difficult in GANs, and most GANs involve constant experimentation parameter modification to achieve this balance. However, since the network of the proposed MsGAN contains multiple generators and multiple confronters, in addition to the parameter optimization between each pair of generators and discriminators, it also involves more complex coordination among multiple generators to obtain better fused images. As a result, this paper proposes to pre-train each generative adversarial network separately, and to then initialize the network with the pre-trained parameters. Finally, joint end-to-end training is carried out to optimize the parameters of each sub-image in a targeted manner.

## 4. Experimental Results and Analysis

In order to verify the superiority of the proposed method, some efficient methods on synthetic and real multi-focus images were employed to conduct extensive comparison experiments, including a multi focus image fusion using high level DWT components and guided filter (DWT) [24], an image fusion algorithm based on spatial frequency-motivated pulse coupled neural networks in the non-subsampled contourlet transform domain (NSCT) [25], a general framework for image fusion based on multi-scale transform and sparse representation (MST-SR) [26], an ensemble of CNN for multi-focus image fusion (ECNN) [27], and a general image fusion framework based on a convolutional neural network (IFCNN) [28]. The parameters in the compared fusion methods were set as recommend by the corresponding authors. For the proposed method, image fusion can also be realized without the discriminative network; in this case, the proposed method without the discriminative network (MsCNN) is equal to an image fusion method based on a multi-scale convolutional neural network. By comparing MsCNN and the proposed method, the necessity of the discriminative network can be proven. In addition, by comparing ECNN and MsCNN, the superiority of the multi-scale decomposition to extract the image features in the proposed method can be proven.

There are two ways to evaluate the fused images by the aforementioned methods: subjective evaluation and objective evaluation. The subjective evaluation is realized by measuring the observer’s visual experience of the fused image, which is greatly influenced by the subjective factors of the observer. In contrast, objective evaluation is obtained by calculating some metrics of fused images, and is more convincing. Therefore, six normalized fusion metrics in [29] were employed to evaluate the fused images more objectively, namely, normalized mutual information ($Q_{MI}$), a fusion metric based on Tsallis entropy ($Q_{TE}$), nonlinear correlation information entropy ($Q_{NCIE}$), gradient-based fusion performance ($Q_{G}$), an image fusion metric based on phase congruency ($Q_{P}$), and the Chen–Blum metric ($Q_{CB}$). Mutual information (MI) in image fusion can measure the mutual dependence of source images and the fused image, and Hossny et al. gave the definition of $Q_{MI}$ to evaluate the information in fused images. Similarly, Tsallis entropy is another divergence measure of the degree between images, and is used to calculate $Q_{TE}$. $Q_{NCIE}$ is a measure of the nonlinear correlation information entropy. Thus, $Q_{MI}$, $Q_{TE}$, and $Q_{NCIE}$ belong to information theory-based metrics. Because the commonly used MI-based metric is sensitive to impulsive noise and its value changes obviously when there is additive Gaussian noise in an image, $Q_{TE}$ and $Q_{NCIE}$ are also employed to compensate for the shortcomings of $Q_{MI}$ and to assess the information entropy of fused images more comprehensively. As another type of assessment metric, $Q_{G}$ and $Q_{P}$ are image feature-based metrics and measure the features transferred from source images to the fused image. In contrast, $Q_{G}$ measures the amount of transferred edge information by the Sobel edge operator, and $Q_{P}$ provides an absolute measure of the transferred image features by the phase congruency. Finally, $Q_{CB}$ is a type of human perception-inspired fusion metric, and there are five steps [30]: extract edge information, partition images into local regions, calculate local regions’ saliency, measure similarity in the local region, and measure global quality.

### 4.1. Implementation Details

The proposed network was trained in a computer whose CPU is an Intel Core i7-6850K with frequency 3.6 GHz. The GPU processor is a NVIDIA RTX 2080Ti with 11 G memory and the operating system is the Ubuntu16.04.

In the training process, all image pairs are resized to 256×256 pixels, randomly flipped horizontally or vertically, and rotated 90, 180, or 270 degrees randomly during training. The network is trained with a learning rate of $1e^{-4}$ and a batch size of 128 for 1000 epochs. To stabilize the training, the learning rate was decayed by $1e^{-1}$ for every 500 epochs.

### 4.2. Synthetic Multi-Focus Image Dataset

Limited by the camera depth of field and registration technology, for multi-focus image fusion, the “standard, well-fused” all-in-focus image needs to be solved and does not exist in the true sense. As a result, there are few publicly available multi-focus image datasets, and it is very challenging to obtain ideal training samples. Considering that the out-of-focus area of a multi-focus image is equivalent to the corresponding focus region multiplied by the blur kernel, we add Gaussian blur kernels with different standard deviation to images in the segmentation dataset, and obtain diverse synthetic multi-focus images, shown in Figure 5. Specially, when there are multiple objects in one image, different blur kernels are added to different areas to simulate the imaging effect of the camera on objects with different depths of field. Although there are tens of thousands of images in the image segmentation dataset, some images with an insignificant difference in the depth of field and a large difference in the object and background scale are removed to better construct a dataset that more closely resembles the real multi-focus images, resulting in about 1200 pairs of source images. Among these, 40% of the pairs are used as the training set, 40% of the pairs are used as the test set, and 20% are used as the validation set. Although the number of images in the synthetic dataset is not large and clipping images into the blocks were not applied to extend the dataset, good fusion results by the proposed method are realized due to the excellent performance of MsGAN.

### 4.3. Compared Experiments

In order to verify the generality and practicability of the proposed method in this paper, the compared experiments conducted on the Lytro dataset are presented in this sub-section. As shown in Figure 6, there are twenty-four sets of real multi-focus images, including four sets of three source images. These images with rich features are left and right focused, and foreground and background focused, making the dataset persuasive enough for multi-focus image fusion. To better visualize the fused images of the Lytro dataset, some typical fused images using different fusion methods are shown in Figure 7, Figure 8 and Figure 9, and some areas are enlarged in green and blue boxes to show the visual contrast effect.

It can be seen that the hand holding the camera in Figure 7a is in focus, whereas the globe and other backgrounds in the mall are out of focus. The scenarios in Figure 7b are completely opposite to those in Figure 7a. When we want to detect the people in the obtained photo showing the hand, some detections will be missed, such as the persons in the enlarged green box of Figure 7b; and the same person can be clearly detected in Figure 7a. Furthermore, the persons in the blue box of Figure 7a, especially the person in the back, are very blurred. By general observation, all the mentioned methods achieved image fusion results and can improve the accuracy of object detection. However, careful comparison of the fused images demonstrates that the fused image by the proposed method has better visual effects and retains more image features.

All the fused images in Figure 8 successfully restored the sunlight refracted shadows on the roof in the blue box, which are lost in Figure 8a. However, by careful comparison, the fused images in Figure 8c,d fail to recover the details in the enlarged green box where a wire rope runs through a small hole. There are also obvious artifacts in the surroundings of “Heart” in Figure 8g. As a summary, the proposed MsGAN performs better at reserving the detailed features and can reconstruct a more natural and realistic fused image.

Figure 9 shows the source and fused images of “Zoo”. In the green box of Figure 9b, the letters on the photo are blurry and out of focus, whereas they are clear in Figure 9a. In addition, the white animal on the left cannot be identified when its texture is blurred, and it could be mistaken for a dog or a lion. By image fusion, the recessed belly of the upper leg is clearly shown in the enlarged blue boxes of the fused images. However, the fused images in Figure 9c,e suffer from the loss of a little structure information. By comparing the fused images in Figure 9, the similar conclusion can be drawn that the fused image using the proposed method is improved and has a satisfactory visual effect, indicating that the proposed method is better at image fusion.

After visually proving the efficiency and effectiveness of the proposed method, the objective evaluation metrics mentioned above were employed to evaluate the proposed method objectively; the average metric values of the fused images of the Lytro dataset are shown in Table 2. For example, the value “0.8860” in the second column and third row is calculated by averaging the values of $Q_{MI}$ of the images fused by DWT. In Table 2, it can be seen that all the values of MsCNN are better than those of ECNN, indicating that the adoption of the multi-scale decomposition is helpful for improving image fusion results. It is difficult to judge the fusion ability of MST-SR and MsCNN by Table 2 alone; however, the values of the proposed method are better than those of MsCNN, strongly indicating the importance of the discriminator network in the proposed method. From Table 2, we can see that the values of image feature-based metrics of the image fused by the proposed method are second and worse than those of MST-SR. However, other values of the image fused by the proposed method are better; the image fused by the proposed method achieves a better value of $Q_{CB}$, which is consistent with the better visual effect of the fused image. By these metric values shown in Table 2, it can be convincingly and objectively demonstrated that the proposed method has better performance in multi-focus image fusion.

## 5. Conclusions

By combing multi-scale decomposition and a generative adversarial network, a new method is proposed in this paper to realize multi-focus image fusion. The source images are decomposed into different sub-images with different feature scales, and then the convolutional neural networks with the same structure and different depths are used to extract the image features complementary to the scale features. Finally, a fused all-in-focus image can be obtained by the competition between the generative network and the discriminant network. The extensive experiments demonstrated that the proposed method achieves better results for multi-focus image fusion.

## Figures and Tables

**Figure 1 entropy-24-00582-f001:**
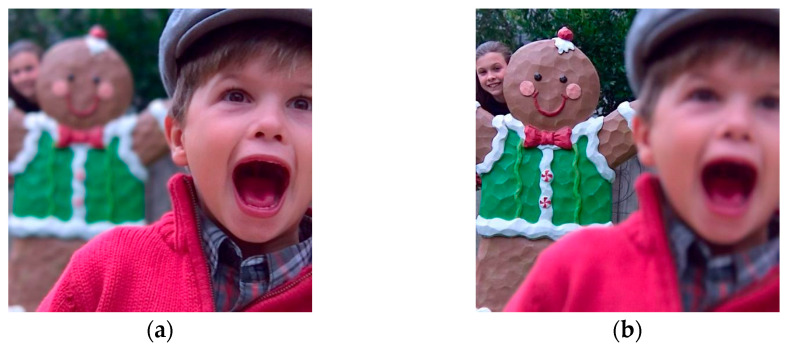
Multi-focus source image: (**a**) right-focused image; (**b**) left-focused image.

**Figure 2 entropy-24-00582-f002:**
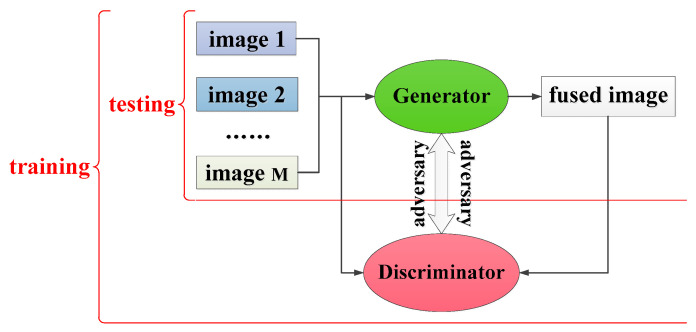
The framework of image fusion based on a GAN.

**Figure 3 entropy-24-00582-f003:**
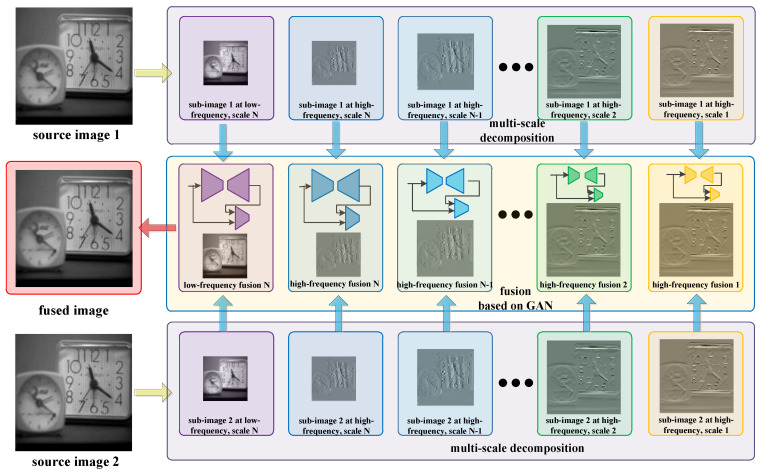
The framework of the multi-focus image fusion based on a multi-scale GAN.

**Figure 4 entropy-24-00582-f004:**
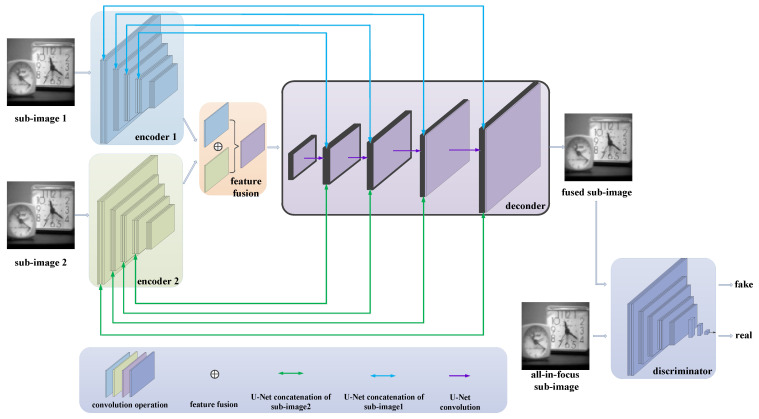
The framework of the proposed sub-image fusion based on a GAN.

**Figure 5 entropy-24-00582-f005:**
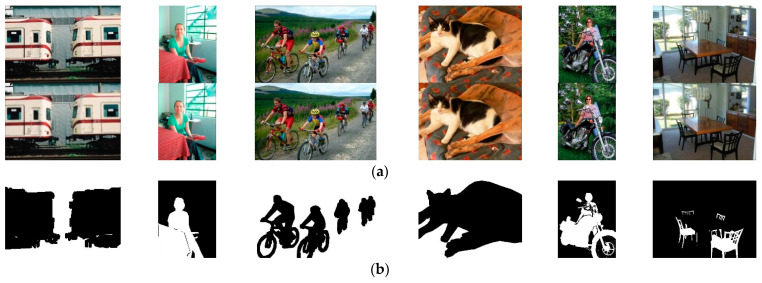
The synthetic multi-focus images: (**a**) multi-focus source image; (**b**) focus area detection map.

**Figure 6 entropy-24-00582-f006:**
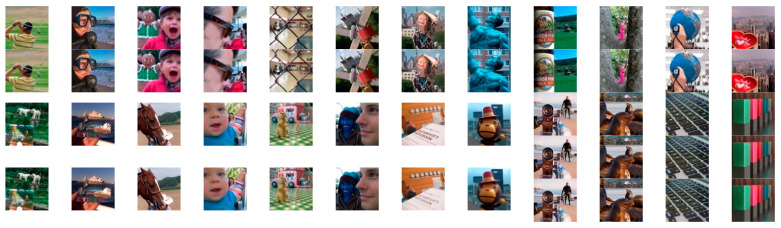
The multi-focus images in the Lytro dataset.

**Figure 7 entropy-24-00582-f007:**
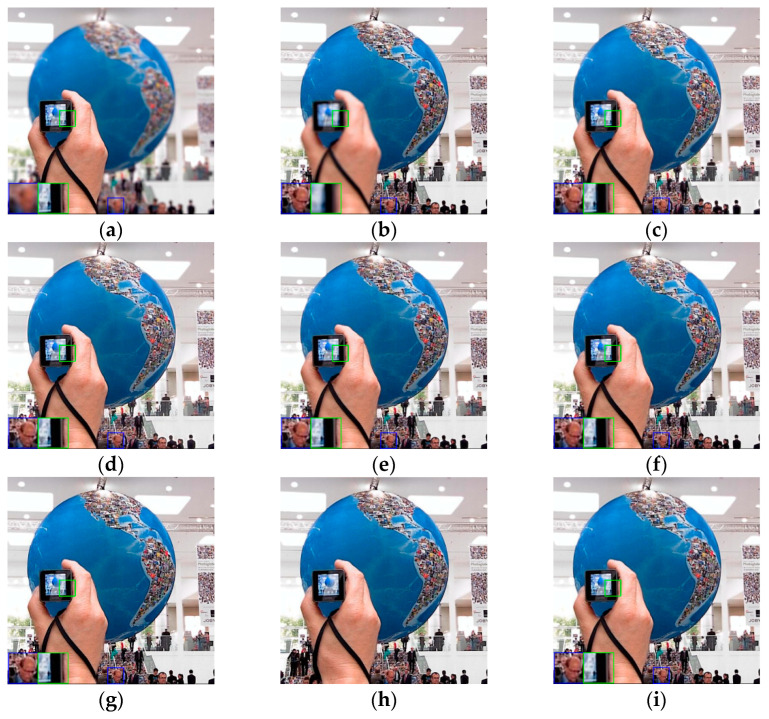
The fused images of “Globe”: (**a**) source image A; (**b**) source image B; (**c**) DWT; (**d**) NSCT; (**e**) MST-SR; (**f**) IFCNN; (**g**) ECNN; (**h**) MsCNN; (**i**) proposed.

**Figure 8 entropy-24-00582-f008:**
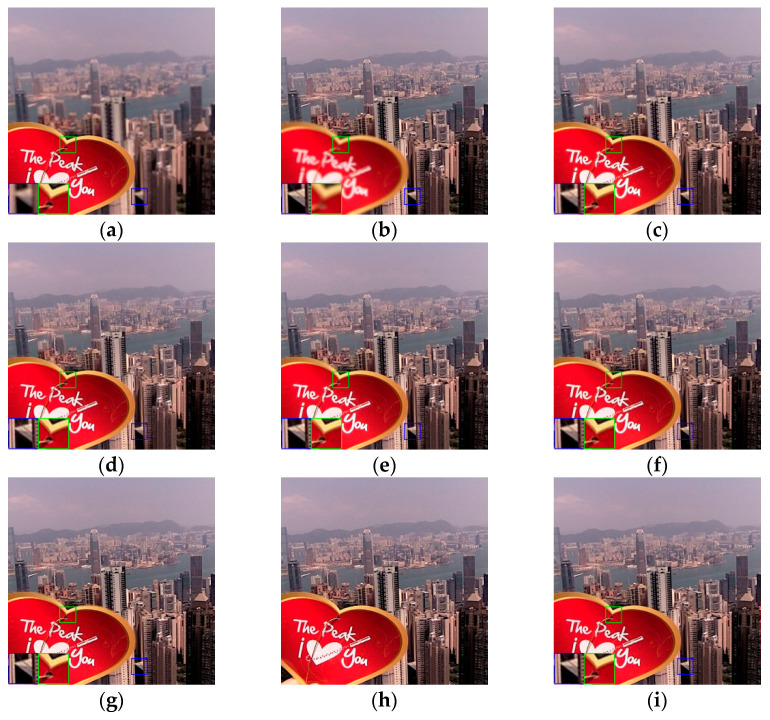
The fused images of “Heart”: (**a**) source image A; (**b**) source image B; (**c**) DWT; (**d**) NSCT; (**e**) MST-SR; (**f**) IFCNN; (**g**) ECNN; (**h**) MsCNN; (**i**) proposed.

**Figure 9 entropy-24-00582-f009:**
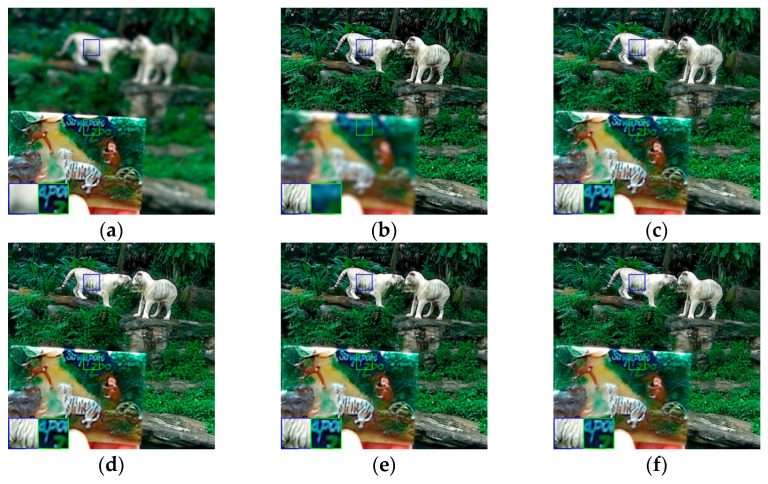
The fused images of “Zoo”: (**a**) source image A; (**b**) source image B; (**c**) DWT; (**d**) NSCT; (**e**) MST-SR; (**f**) IFCNN; (**g**) ECNN; (**h**) MsCNN; (**i**) proposed.

**Table 1 entropy-24-00582-t001:** The details of the generator and discriminator.

		Layer	Convolution	Normalization	Activation
G	Encoder	En_1	Conv(3,64,3,1,1)	BatchNorm2d	Leaky ReLU
			Conv(64,64,3,1,1)	BatchNorm2d	Leaky ReLU
		En_2	Conv(64,128,4,2,1	BatchNorm2d	Leaky ReLU
			Conv(128,128,3,1,1)	BatchNorm2d	Leaky ReLU
		En_3	Conv(128,256,4,2,1)	BatchNorm2d	Leaky ReLU
			Conv(256,256,3,1,1)	BatchNorm2d	Leaky ReLU
		En_4	Conv(256,512,4,2,1)	BatchNorm2d	Leaky ReLU
			Conv(512,512,3,1,1)	BatchNorm2d	Leaky ReLU
		En_5	Conv(512,512,4,2,1)	BatchNorm2d	Leaky ReLU
			Conv(512,512,3,1,1)	BatchNorm2d	Leaky ReLU
	Feature Fusion	FF	Concat(f1, f2)	-	-
	Decoder	De_1	ConvT (512,512,4,2,1)	-	Leaky ReLU
			Conv(512, 512,3,1,1)*2	BatchNorm2d	Leaky ReLU
		De_2	ConvT (512,256,4,2,1)	-	Leaky ReLU
			Conv(256,256,3,1,1)*2	BatchNorm2d	Leaky ReLU
		De_3	ConvT (256,128,4,2,1)	-	Leaky ReLU
			Conv(128,128,3,1,1)*2	BatchNorm2d	Leaky ReLU
		De_4	ConvT(128,64,4,2,1)	-	Leaky ReLU
			Conv(64,64,3,1,1)*2	BatchNorm2d	Leaky ReLU
		De_5	Conv(64,3,3,1,1)	-	Tanh
D		D_1	Conv(3,64,3,1,1)	BatchNorm2d	Leaky ReLU
			Conv(64,64,3,1,1)	BatchNorm2d	Leaky ReLU
		D_2	Conv(64,128,4,2,1)	BatchNorm2d	Leaky ReLU
			Conv(128,128,3,1,1)	BatchNorm2d	Leaky ReLU
		D_3	Conv(128,256,4,2,1)	BatchNorm2d	Leaky ReLU
			Conv(256,256,3,1,1)	BatchNorm2d	Leaky ReLU
		D_4	Conv(256,512,4,2,1)	BatchNorm2d	Leaky ReLU
			Conv(512,512,3,1,1)	BatchNorm2d	Leaky ReLU
		D_5	Conv(512,512,4,2,1)	BatchNorm2d	Leaky ReLU
			Conv(512,512,3,1,1)	BatchNorm2d	Leaky ReLU

**Table 2 entropy-24-00582-t002:** The average metric values of the fused images of Lytro dataset.

	Information Theory Based Metrics			Image Feature Based Metrics		Human Perception Inspired Fusion
	$Q_{MI}$	$Q_{TE}$	$Q_{NCIE}$	$Q_{G}$	$Q_{P}$	$Q_{CB}$
DWT [24]	0.8860	0.3747	0.8274	0.6483	0.7836	0.7073
NSCT [25]	0.9521	0.3782	0.8317	0.6310	0.7689	0.7238
MST-SR [26]	0.9594	0.3852	0.8309	0.6905	0.8247	0.7515
ECNN [27]	0.8877	0.3796	0.8272	0.6396	0.7842	0.7129
IFCNN [28]	0.8580	0.3759	0.8258	0.6195	0.7665	0.6868
MsCNN	0.9602	0.3829	0.8309	0.6503	0.7877	0.7504
Proposed	0.9945	0.3861	0.8329	0.6727	0.8028	0.7654

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
