# Peer review of "Multi-Focus Image Fusion Based on Multi-Scale Generative Adversarial Network"

_entropy, 2022, doi:10.3390/e24050582_

Round 1

Reviewer 1 Report

  • the work should be checked by a native English speaker because there a few expressions to correct. For example, in page 3 (line 109) "and the computation consume", in line 131 "the related work", in line 287 "the experimental results..."
  • In line 292-295 the authors mention  the use of "general frameworks"  perhaps they should write "application of a general framework..."
  • In table 2, I suppose that the values of the different Q are averaged on the values of different examples (24), it should be clarified

Author Response

Dear Reviewer,

Thank you for the efforts made in processing this submission and we are particularly grateful to you for the constructive comments and suggestions which help us improve the quality of this manuscript. We feel very sorry for problems in the prior manuscript. And we have carefully revised this manuscript according the comments, and the changes are highlighted in yellow color in the revised manuscript. The point-by-point reply can be found in the following pages.

Comment (1): the work should be checked by a native English speaker because there a few expressions to correct. For example, in page 3 (line 109) "and the computation consume", in line 131 "the related work", in line 287 "the experimental results..."

Response: Thanks for your careful reading and constructive suggestions. We feel very sorry for our errors, and we have checked and corrected them in the revised manuscript. In addition, we asked a native professor for help to polish our manuscript to make the expression more typical and fluent.

Comment (2): In line 292-295 the authors mention the use of "general frameworks" perhaps they should write "application of a general framework..."

Response: Thank you for your valuable comments. In fact, we employed some compared image fusion methods and cited the corresponding references. And the authors in these references used “general frameworks” as part of their titles; therefore, “general frameworks” is used in our manuscript. Thank you again for your patience and hard work on our manuscript.

Comment (3): In table 2, I suppose that the values of the different Q are averaged on the values of different examples (24), it should be clarified

Response: Thank you for your constructive comments, and we feel very sorry for our unclear statement. The title of Table 2 is “The average metric values of the fused images of Lytro dataset”, indicating that the values in this table are averaged on the values of fused images of Lytro dataset with 24 group images. To make it clearer and more understandable, in the revised manuscript, we have added some descriptions as “For example, the value “0.8860” in the second column and third row is calculated by averaging the values of  QMI of fused images by DWT.”, and the changes are highlighted in yellow. Finally, thank you very much for providing the valuable suggestions, and we feel great thanks for your professional review on our manuscript.

Reviewer 2 Report

This paper proposes a multi-focus image fusion method based on Wavelet decomposition and adversarial learning. Although the proposed method is compared with several previous methods, there are several weaknesses in the current manuscript. The most critical point is that ablation studies are not conducted for demonstrating the effectiveness of the proposed feature fusion method and adversarial learning on the multi-focus fusion problem. In my opinion, major revision is required to improve the quality of the manuscript, and the reviewer’s comments are as follows.

  1. The introduction section is well-written; however, a significant improvement on the proposed method section is required.
  2. There are several grammatical errors in most equations including omitted commas and periods. Eq.(1) and Eq.(2) need to be modified into (1) and (2) in lines 191 and 196.
  3. In (1)~(3), min and max are omitted on the righthand side of the equations.
  4. The quality of Figure 2 needs to be improved.
  5. In Figure 4, the model does not contain any layer corresponding to the color which indicates U-Net connection and convolution. Moreover, the term “feature fusion” in Figure 4 is ambiguous.
  6. In experiments, the authors have to show the metric values with and without the feature fusion module. Moreover, the metric values without the discriminative network are also required.
  7. The authors have to describe more details about the training, validation, and test sets.
  8. In Table 2, reference numbers are omitted for the previous method.

Author Response

Dear Reviewer,

Thank you for the efforts made in processing this submission and we are particularly grateful to you for the constructive comments and suggestions which help us improve the quality of this manuscript. We feel very sorry for problems in the prior manuscript. And we have carefully revised this manuscript according the comments, and the changes are highlighted in yellow color in the revised manuscript. The point-by-point reply can be found in the attachment.

Round 2

Reviewer 2 Report

The authors conducted major revision, and all of the reviewer's concerns are resolved. I believe that this manuscript meets the minimum qualification for publication in this journal.